# Generation of Artificial Blastoids Combining miR-200-Mediated Reprogramming and Mechanical Cues

**DOI:** 10.3390/cells13070628

**Published:** 2024-04-04

**Authors:** Georgia Pennarossa, Sharon Arcuri, Fulvio Gandolfi, Tiziana A. L. Brevini

**Affiliations:** 1Laboratory of Biomedical Embryology and Tissue Engineering, Department of Health, Animal Science and Food Safety and Center for Stem Cell Research, Università degli Studi di Milano, 20133 Milan, Italy; sharon.arcuri@unimi.it; 2Department of Agricultural and Environmental Sciences-Production, Landscape, Agroenergy, Università degli Studi di Milano, 20133 Milan, Italy; fulvio.gandolfi@unimi.it

**Keywords:** blastoids, cellular reprogramming, ICM-like spheroids, miR-200 family, TR-like cells

## Abstract

In vitro-generated blastocyst-like structures are of great importance since they recapitulate specific features or processes of early embryogenesis, thus avoiding ethical concerns as well as increasing scalability and accessibility compared to the use of natural embryos. Here, we combine cell reprogramming and mechanical stimuli to create 3D spherical aggregates that are phenotypically similar to those of natural embryos. Specifically, dermal fibroblasts are reprogrammed, exploiting the miR-200 family property to induce a high plasticity state in somatic cells. Subsequently, miR-200-reprogrammed cells are either driven towards the trophectoderm (TR) lineage using an ad hoc induction protocol or encapsulated into polytetrafluoroethylene micro-bioreactors to maintain and promote pluripotency, generating inner cell mass (ICM)-like spheroids. The obtained TR-like cells and ICM-like spheroids are then co-cultured in the same micro-bioreactor and, subsequently, transferred to microwells to encourage blastoid formation. Notably, the above protocol was applied to fibroblasts obtained from young as well as aged donors, with results that highlighted miR-200′s ability to successfully reprogram young and aged cells with comparable blastoid rates, regardless of the donor’s cell age. Overall, the approach here described represents a novel strategy for the creation of artificial blastoids to be used in the field of assisted reproduction technologies for the study of peri- and early post-implantation mechanisms.

## 1. Introduction

Understanding the main mechanisms underlying the specific events taking place during the early phase of embryogenesis represents a “critical step” towards developing novel treatments for infertility and developmental disorders. In addition, the study of the peri-implantation period in humans should be useful to better characterize the role of heritable epigenetics and may help in the production of terminally functional differentiated cells to be used in regenerative medicine [1]. Nonetheless, the use of human embryos is hindered by in vivo environment inaccessibility as well as by material paucity and the ethical and legal issues involved [2]. To overcome these limitations, several attempts have been directed towards the generation of in vitro models able to mimic embryogenesis, and, in recent years, a set of novel stem cell-based embryo models, defined blastoids [3,4,5,6,7,8,9,10], have seen the light [2]. Blastoids represent ideal tools to complement research on natural embryos since they contain both embryonic and extraembryonic cell types and can partially mimic early embryonic development in vitro. Furthermore, they can be generated in high quantities with similar genetic makeups and are amenable to experimental manipulation, including genetic screens, thus providing promising alternatives to the use of human embryos [2,11].

To date, different approaches based on the self-aggregation of various stem cell types have been described for the production of embryo-like structures [3,4,5,6,7,8,9,10]. In this context, we have recently reported the possibility of generating 3D multicellular spherical structures that are remarkably similar to natural embryos. We start with easily obtainable terminally differentiated cells and combine the use of epigenetic cues to erase the original cell phenotype [12,13] and mechanical stimuli to ease cell aggregation and encourage pluripotency [14,15]. The epigenetic modifier used in that experiment was the well-known eraser agent 5-azacytidine (5-aza-CR), which is able to induce both direct and indirect demethylating events [12]. Interestingly, it has been demonstrated that its direct effect is driven by the ten-eleven translocation (TET) family enzymes, which, in turn, activate the family of miR-200 [16,17]. Indeed, the latter is involved in the removal of epigenetic blocks and in the re-activation of previously silenced genes during cellular reprogramming [18,19,20,21], thus promoting pluripotency and helping fibroblasts to overcome the mesenchymal–epithelial transition (MET) barrier [22]. In addition, it has been recently demonstrated that the use of miR-200s ameliorates senescence in aged cells [16,17], which, however, maintain their phenotype, thus undergoing “partial reprogramming”, also defined as “age reprogramming” [23].

Based on this observation, in the present work, we investigate whether the combination of miR-200 with mechanical cues would allow successful “developmental reprogramming”, with cell de-differentiation to a high plasticity state and the generation of blastoid structures, starting from easily accessible terminally differentiated fibroblasts isolated from young and aged individuals (Figure 1). To this purpose, cells belonging to the two experimental groups were transfected with miR-200s to induce a high plasticity state. Subsequently, miR200-reprogrammed young and aged cells were either driven towards the TR lineage by using an ad hoc induction protocol [24] or encapsulated into polytetrafluoroethylene (PTFE) micro-bioreactors to support tridimensional cell aggregation and encourage pluripotency [14,15,25]. Lastly, the generated TR-like cells and ICM-like spheroids were co-cultured in the same micro-bioreactor and then transferred to microwells to encourage differentiation and favor blastoid generation [14].

## 2. Materials and Methods

All reagents were obtained from Thermo Fisher Scientific (Milan, Italy), unless otherwise indicated.

### 2.1. Ethical Statement

Human cell lines (GM02674, GM00495, GM08402, GM01706, GM00731, AG09602) were purchased from the NIGMS Human Genetic Cell Repository at the Coriell Institute for Medical Research (Camden, NJ, USA). This study did not involve human participants; therefore, ethical approval was not required.

### 2.2. Culture of Human Skin Fibroblasts

Fibroblasts obtained from 3 young individuals (29–32 years old) and from 3 aged donors (82–96 years old) were grown in a fibroblast standard culture medium (FCM) consisting of Eagle’s minimum essential medium supplemented with 15% not heat-inactivated fetal bovine serum (FBS), 2 mM glutamine (Sigma-Aldrich, Milan, Italy), and a 1% antibiotic/antimycotic solution (Sigma-Aldrich, Milan, Italy). Cells were cultured in 5% CO_2_ at 37 °C and passaged twice a week at ratios of 1:3 (young) and 1:2 (aged). All experiments were carried out using all 6 human fibroblast cell lines at passages between 6 and 8 at least three times in triplicate.

### 2.3. Fibroblast Transfection with Synthetic miR-200s

Fibroblasts isolated from young and aged individuals were seeded into 4-well multidishes (Nunc, Milan, Italy) pre-treated with 0.1% gelatin (Sigma-Aldrich, Milan, Italy) at a density of 2.1 x 10^4^ cells/cm^2^. At 24 h after seeding, the FCM was removed, cells were washed in PBS and transfected with 100 nM of miR-200b/c (predesigned mirVANA Mimics) using RNAi-MAX in Opti-MEMTM medium, following the manufacturer’s instructions, and incubated for 48 h in 5% CO_2_ at 37 °C. Concentrations and times of exposure were selected according to our previous work [16].

### 2.4. Generation of TR-like Cells

At the end of synthetic miR-200b/c treatment, cells were incubated in embryonic stem cell (ESC) medium consisting of DMEM-low glucose: HAM’S F10 (1:1), 5% FBS, 10% K.O. serum, 2 mM glutamine (Sigma-Aldrich, Milan, Italy), 0.1 mM β-mercaptoethanol (Sigma-Aldrich, Milan, Italy), nucleoside mix, 1% non-essential amino acids, 1000 IU/mL ES-growth factor (LIF, Chemicon, Milan, Italy), and 5 ng/mL b-FGF (R&D System, Milan, Italy) [26] for 3 h in 5% CO_2_ at 37 °C. TR induction was then induced by culturing cells in mouse embryonic fibroblast (MEF)-conditioned medium [24] supplemented with 10 ng/mL bone morphogenetic protein 4 (BMP4, Sigma-Aldrich, Milan, Italy), 1 µM activin/nodal signaling inhibitor (A83-01, Sigma-Aldrich), and 0.1 µM basic fibroblast growth factor (FGF2)-signaling inhibitor (PD173074, Sigma-Aldrich, Milan, Italy) [24,26,27] at 37 °C in low O_2_ condition (5% O_2_, 5% CO_2_, 90% N2 atmosphere) for 11 days. The culture medium was changed every other day.

### 2.5. Production of ICM-like Spheroids

At the end of synthetic miR-200b/c treatment, cells were resuspended in ESC medium and encapsulated in PTFE (Sigma-Aldrich, Milan, Italy) micro-bioreactors for 24 h [15]. A PTFE powder bed with a particle size of 1 μm (Sigma-Aldrich, Milan, Italy, 430935) was prepared inside a culture dish (Sarstedt, Milan, Italy), and 1 × 10^4^ cells/30 μL of ESC medium was dispensed on it. Micro-bioreactors were generated by gently rotating in a circular motion the petri dish, transferred to a new petri dish, and cultured in 5% CO_2_ at 37 °C using a humidified chamber to avoid dehydration.

### 2.6. Creation of Blastoids

Blastoids were generated by assembling TR-like cells and ICM-like spheroids, as previously described [14]. More in detail, the TR-like cells differentiated for 11 days were detached from culture dishes and resuspended in G1-PLUS medium (Vitrolife, Massa, Italy) at 3 × 10^4^ cells/30 μL concentration. In parallel, ICM-like spheroids were collected from the micro-bioreactors and transferred into a drop of G1-PLUS medium (Vitrolife, Massa, Italy). A 40 μL drop containing 3 × 10^4^ TR-like cells (30 μL) and one single ICM-like spheroid (10 μL) was dispensed onto a PTFE powder bed to create new micro-bioreactors that were cultured for 2 days at 37 °C in 5% CO_2_. The generated blastoids were collected and cultured into non-adherent microwells (AggreWellTM, Stemcell Technologies, Basel, Switzerland) in G2-PLUS medium (Vitrolife, Massa, Italy). After 24 h, 10^5^ PODS Activin A (Cell Guidance Systems, Cambridge, UK) were added to 150 μL of G2-PLUS medium (Vitrolife, Massa, Italy), and blastoids were maintained for an additional 4 days in 5% CO_2_ at 37 °C.

### 2.7. Morphological and Morphometric Evaluations

Cell morphology was monitored daily using an Eclipse TE200 inverted microscope (Nikon, Firenze, Italy), connected to a Digital Sight camera (Nikon, Firenze, Italy). ICM-like spheroid and blastoid morphometric evaluations were carried out by acquiring pictures with NIS-Elements Software (Version 4.6; Nikon, Firenze, Italy). Spheroid diameters were then measured using ImageJ software (ImageJ software version 1.53j).

### 2.8. Gene Expression Analysis

RNA was extracted using the TaqManGene Expression Cells-to-CT kit following the manufacturer’s instructions. DNase I (1:100) was added to the lysis solution. Quantitative real-time PCR was carried out using predesigned gene-specific primers and probe sets (TaqManGene Expression Assays, Table 1). The GAPDH and ACTB genes were selected as reference genes. Target gene analysis was carried out using the CFX96 Real-Time PCR detection system (Bio-Rad Laboratories, Milan, Italy) and the CFX Manager software (Bio-Rad Laboratories version 3.1).

### 2.9. Blastoid Cell Separation

Single blastoids were separately analyzed. More in detail, each blastoid was dissociated to a single cell suspension by a double enzymatic digestion with collagenase IV (300 U/mL, Sigma-Aldrich, Milan, Italy) for 30 min and trypsin–EDTA solution (Sigma-Aldrich, Milan, Italy) for 20 min, followed by mechanical dissociation by pipetting. Cell suspension was filtered with a 30 µm nylon mesh (Pre-Separation Filters, 30 μm, # 130-041-407, Miltenyi Biotec, Bologna, Italy) and centrifuged at 300× *g* for 5 min. Supernatants were disposed of, and trophoblast cell surface antigen 2 (TROP2)+ cells were isolated using the magnetic-activated cell sorting (MACS, Miltenyi Biotec, Bologna, Italy) protocol, following the manufacturer’s instructions. TROP2− cells were isolated by applying the same protocol and collecting the flow-through. The two cell populations obtained were subjected to gene expression analysis.

### 2.10. Immunocytochemical Analysis

Cells and blastoids were fixed in 4% paraformaldehyde for 20 min, rinsed in PBS, permeabilized with 0.5% Triton X-100 (Sigma-Aldrich, Milan, Italy) for 30 min, and incubated with a blocking solution containing 10% goat serum (Sigma-Aldrich, Milan, Italy) for 30 min. ICM-like spheroids were dissociated and attached to slides using a cytocentrifuge, Cytospin 4 (Thermo Shandon, Milan, Italy), before immunocytochemical staining. Primary antibodies for OCT4 (1:200, Chemicon, Milan, Italy, ab3209), KRT19 (1:200, Abcam, Cambridge, UK, ab76539), and CDX2 (1:50, Santa Cruz Biotechnology, Milan, Italy, sc-166830) were incubated overnight at +4 °C. Samples were then washed with PBS and incubated with the appropriate secondary antibodies (Alexa Fluor) for 45 min at room temperature using a 1:250 dilution. Nuclei were counterstained with 4′,6-diamidino-2-phenylindole (DAPI, Sigma-Aldrich, Milan, Italy). At the end of the immunostaining procedure, cells were analyzed under an Eclipse E600 microscope (Nikon, Firenze, Italy) equipped with a digital camera (Nikon, Firenze, Italy); blastoids were transferred, mounted to glass slides, and visualized under an Eclipse E600 microscope (Nikon, Firenze, Italy) equipped with a digital camera (Nikon, Firenze, italy). Images were acquired using NIS-Elements Software (Version 4.6; Nikon).

### 2.11. Statistical Analysis

Statistical analysis was carried out using the Student’s *t*-test (SPSS 19.1; IBM). Results were reported as mean ± standard deviation (SD). Differences of *p* ≤ 0.05 were considered significant and were indicated with different superscripts.

## 3. Results

### 3.1. Induction of High Plasticity Using miR-200-Mediated Reprogramming

After miR-200 transfection, fibroblasts isolated from both young and aged individuals displayed considerable phenotype changes. The typical elongated shape, detectable in control fibroblasts (T0, Figure 2A), switched to a stem-cell-like morphology. Cells became smaller with larger nuclei, granular and vacuolated cytoplasm, and rearranged in a reticular pattern, tending to form distinguishable aggregates (miR-200, Figure 2A). This was accompanied by the onset of the main pluripotency-related genes, namely OCT4, NANOG, REX1, and SOX2, which were not expressed in control fibroblasts (T0, Figure 2B). It is interesting to note that the transcription levels of all the genes analyzed were statistically comparable between the two experimental groups (young and aged, Figure 2B). These molecular data were also supported by immunocytochemical analysis, demonstrating OCT4 positivity in both young and aged cells transfected with miR-200 (young miR200 and aged miR200, Figure 2C).

### 3.2. Generation of TR-like Cells from miR-200-Reprogrammed Cells

After 11 days of TR induction, both young and aged cells showed a tight adherent epithelial morphology, exhibiting a round or ellipsoid shape, round nuclei, and well-defined borders, regardless of the donor age (young and aged, Figure 3A). Consistent with the acquisition of the typical TR-like phenotype, cells actively expressed the mature TR-related markers, namely GCM1, KRT19, PGF, CYP11A1, CGA, ESRRB, CGB, and HSD17B1, which were absent in control fibroblasts (T0, Figure 3B). Interestingly, the expression values detected in TR-like cells were comparable in young and aged cells (miR-200, Figure 3B). Immunocytochemical studies confirmed the molecular results and showed KRT19 immunopositivity in both young and aged cells transfected with miR200 and differentiated towards the trophectoderm lineage (Figure 3C).

### 3.3. Creation of ICM-like Spheroids from miR-200-Reprogrammed Cells

After miR-200 transfection, young and aged cells, encapsulated in PTFE micro-bioreactors, aggregated in 3D spherical structures (young and aged, Figure 4A). Consistently, the generated ICM-like structures actively express the main pluripotency-related genes OCT4, NANOG, REX1, and SOX2, originally absent in untreated cells (T0, Figure 4B). This was also confirmed by the OCT4 immunopositivity of cells isolated from young and aged ICMs (young and aged, Figure 4C).

### 3.4. Creation of Blastoids by Assembling TR-like Cells and ICM-like Spheroids Obtained from miR-200-Reprogrammed Cells

After 2 days of co-culture into PTFE micro-bioreactors, TR-like cells and ICM-like spheroids self-organized into single 3D spheroids (young and aged, Figure 5A), and, at the end of a 7-day culture period in microwells, they generated blastoids with a uniform round shape (young and aged, Figure 5B) and a size ranging from 100 to 200 µm, regardless of the donor age (young and aged, Figure 5C). Morphometric evaluation demonstrated that 79.67 ± 10.23% of the 3D spherical structures obtained using young cells displayed a diameter of 151–200 μm and 20.33 ± 7.25% of 100–150 μm (young, Figure 5C). Similarly, 84.39 ± 12.56% of blastoids generated with cells isolated from aged individuals showed a diameter of 151–200 μm and 15.61 ± 6.45% of 100–150 μm (aged, Figure 5C). Immunostaining highlighted CDX2+ cells confined to the outer layer of the generated blastoids, while OCT4+ cells were localized in the inner compartment (Figure 5D). Furthermore, molecular analyses indicated that TROP2− cells were shown to actively express the pluripotency-related genes OCT4, NANOG, REX1, and SOX2, while TROP2+ cells transcribed the TR-related markers GCM1, KRT19, PGF, CYP11A1, CGA, ESRRB, CGB, and HSD17B1 (Figure 5E).

## 4. Discussion

The successful generation of in vitro blastocyst-like structures starting from embryonic stem cells (ESCs), induced pluripotent stem cells (iPSCs), or epigenetically erased cells has been recently demonstrated [5,7,10,14,28,29,30]. In the present manuscript, we describe a new approach that combines miR-200-mediated reprogramming ability with mechanical stimuli to create in vitro 3D spherical structures that are phenotypically similar to natural embryos. To our knowledge, this strategy has not been applied before for the creation of blastocyst-like structures. In addition to its novelty, this method offers the advantage of easily accessible cells as starting material and avoids the use of any retroviral and/or lentiviral vectors, as well as the insertion of transgenes. The significant improvement offered by the protocol described is also represented by the ease of performance that involves a simple transfection of miR-200s to induce a high permissivity state in adult dermal fibroblasts, obtained either from young or aged donors. Significant changes are evident in both groups, with the acquisition of morphological and molecular features distinctive of pluripotent cells. In particular, the typical fibroblast elongated shape changes to a round or oval one, and cells decrease in size, with granular, vacuolated cytoplasm rearranging in distinguishable aggregates, regardless of the cell donor’s age. All these aspects closely resemble the typical round morphology previously described for human ESCs [31] and iPSCs [32], which form compact colonies with distinct borders and well-defined edges [31,32,33,34]. Consistent with what was recently demonstrated in cells reprogrammed with miR-200s [16], both young and aged transfected fibroblasts show larger nuclei with less cytoplasm, a feature generally associated with the relaxed and accessible chromatin structure typically described in high-plasticity cells [35,36,37,38]. All this indicates miR-200′s ability to encourage properties previously observed in both native and induced pluripotent cells. The switch towards a pluripotent phenotype is also corroborated by the molecular studies that demonstrate the onset of OCT4, NANOG, REX1, and SOX2 transcription. These observations confirm and further expand previous studies demonstrating that miR-200 overexpression results in ESC self-renewal and the induction of Nanog transcription while inhibiting embryoid body formation and repressing the expression of ectoderm-, endoderm-, and mesoderm-related markers [39,40]. This is coherent with the miR-200 family’s ability to remove epigenetic blocks and help fibroblasts overcome the MET barrier during cellular reprogramming [22].

Interestingly, the development of 3D culture systems that support robust long-term and feeder-free self-renewal of pluripotent cells has been recently described and demonstrated to boost and stabilize high plasticity [15,25,41,42]. In line with these findings, encapsulation of miR-200-reprogrammed cells in PTFE micro-bioreactors coax cells to assemble in multicellular 3D spheroids reminiscent of ICM-like morphology and steadily transcribing for the main pluripotency-related genes.

On the other hand, taking advantage of the acquired high plasticity state, miR-200-reprogrammed cells can be driven towards the TR lineage using an ad hoc induction protocol that favors the acquisition of a tight adherent epithelial morphology with a round shape and well-defined borders in both young and aged cells. This is consistent with previous observations that described the differentiation of human iPSCs and epigenetically converted cells towards the TR lineage with the same induction cocktail, containing BMP4, activin/nodal-, and FGF2-signaling inhibitors [24,27,43,44,45,46,47,48,49,50,51,52], and confirms that, when present at adequate concentrations, these molecules are able to coax cells unidirectionally into the TR phenotype [27,49,53,54]. The activation of the molecular pathways distinctive of the newly acquired differentiation is also demonstrated by the active transcription of mature TR-related markers, such as GCM1, KRT19, PGF, CYP11A1, CGA, ESRRB, CGB, and HSD17B1, as well as by KRT19 immunopositivity which was detected in all TR-like cells, regardless of the donor’s age.

Assembly of ICM-like spheroids and TR-like cells in a confined common micro-environment allows the generation of single 3D spherical structures, formed by the two cell components, and encourages the generation of blastoids that display a uniform round shape and diameters statistically comparable in young and aged cells. It is interesting to note that around 80% of the generated structures exhibit a diameter ranging from 151 to 200 μm that well fits with the average reported values of natural blastocysts. We think this is an important point since, together with morphological similarity, measurements are considered a key aspect among the parameters and criteria currently used to define human blastoid models [5,14]. In addition, the localization of cells within the spheroids, demonstrated by the immunostaining studies showing CDX2+ cells localized externally, surrounding the generated blastoids, and OCT4+ cells closely assembled within them, further indicate that the newly formed spheroids display a cell lineage segregation similar to natural blastocysts and that this feature is maintained both in young as well as in aged blastoids. This is further confirmed by the molecular data obtained after sorting blastoid-derived cells with the surface marker TROP2, which highlights a transcription pattern distinctive of the two coexisting components, the TR and the ICM.

The approach presented in this manuscript offers the possibility of easily obtaining artificial blastoids that can find useful application in the study of peri- and early post-implantation mechanisms as well as for the further understanding of early developmental processes. It also offers the advantages of scalability, accessibility, limited variables, and direct manipulation. In the future, these newly generated structures could also be useful in identifying therapeutic targets as well as supporting preclinical modeling, offering an ethical alternative to the use of natural ones.

## Figures and Tables

**Figure 1 cells-13-00628-f001:**
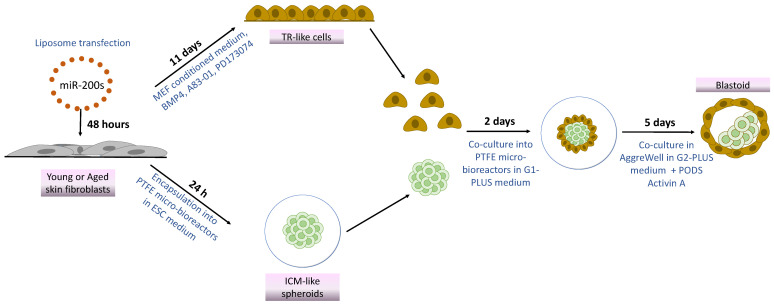
Scheme illustrating blastoid generation from young and aged skin fibroblasts using miR-200s and mechanical cues.

**Figure 2 cells-13-00628-f002:**
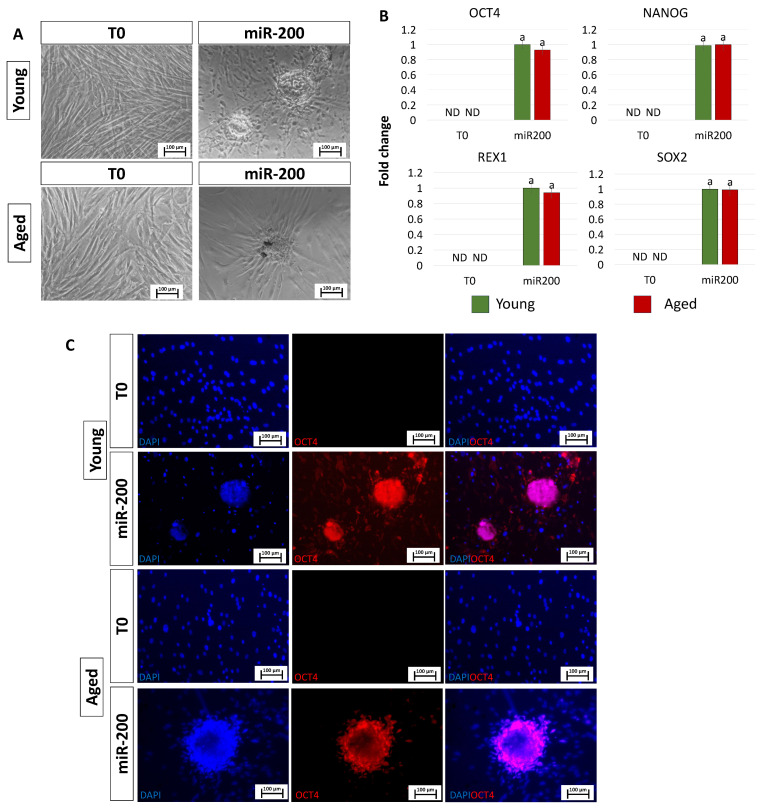
Induction of high plasticity through miR-200-mediated reprogramming: (**A**) After miR-200b/c transfection, fibroblasts isolated from young and aged individuals lost their typical elongated shape (T0) and became smaller in size with granular, vacuolated cytoplasm, larger nuclei, and formed distinguishable aggregates (miR-200) (scale bars: 100 µm). (**B**) Transcription levels for the pluripotent-related genes OCT4, NANOG, REX1, and SOX2 in young (green bars) and aged (red bars) untreated fibroblasts (T0) and in fibroblasts transfected with miR-200 (miR-200). Gene expression is presented with the highest expression set to 1 and all others relative to this. Different superscripts indicate significant differences (*p* < 0.05). (**C**) Immunostainings show immunopositivity of miR-200-reprogrammed young and aged cells for the pluripotency-related marker OCT4. Nuclei are stained blue (scale bars: 100 μm).

**Figure 3 cells-13-00628-f003:**
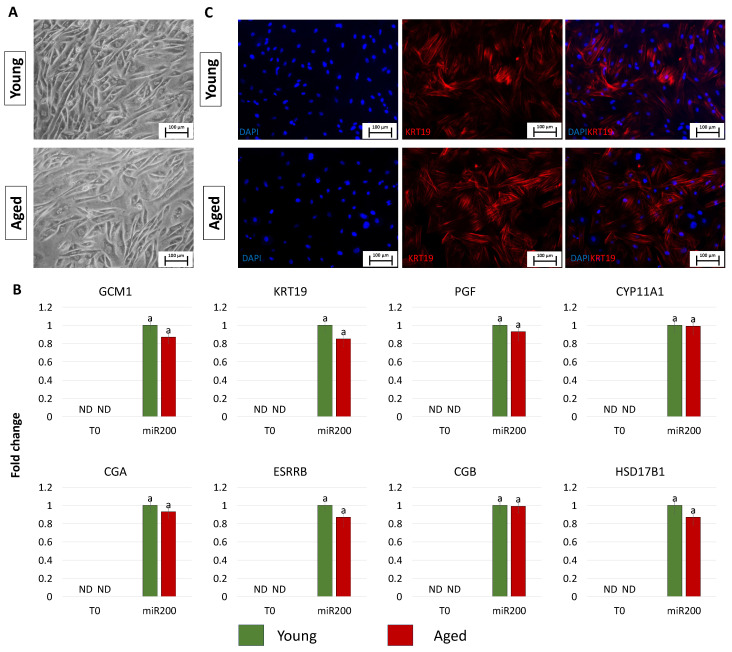
Generation of TR-like cells from miR-200-reprogrammed cells: (**A**) At day 11 of trophoblast induction, young and aged cells acquired a tight adherent epithelial morphology, exhibiting a round or ellipsoid shape with round nuclei and well-defined borders (scale bars: 100 µm). (**B**) Transcription levels for TR-related genes (GCM1, KRT19, PGF, CYP11A1, CGA, ESRRB, CGB, HSD17B1) in untreated young (green bars) and aged (red bars) fibroblasts (T0) and at day 11 of trophoblast induction (miR200). Gene expression levels are reported, with the highest expression set to 1 and all others relative to this. Different superscripts indicate significant differences (*p*  <  0.05). (**C**) Immunostainings show young and aged cell positivity for the TR marker KRT19 (red). Nuclei are stained blue (scale bars: 100 μm).

**Figure 4 cells-13-00628-f004:**
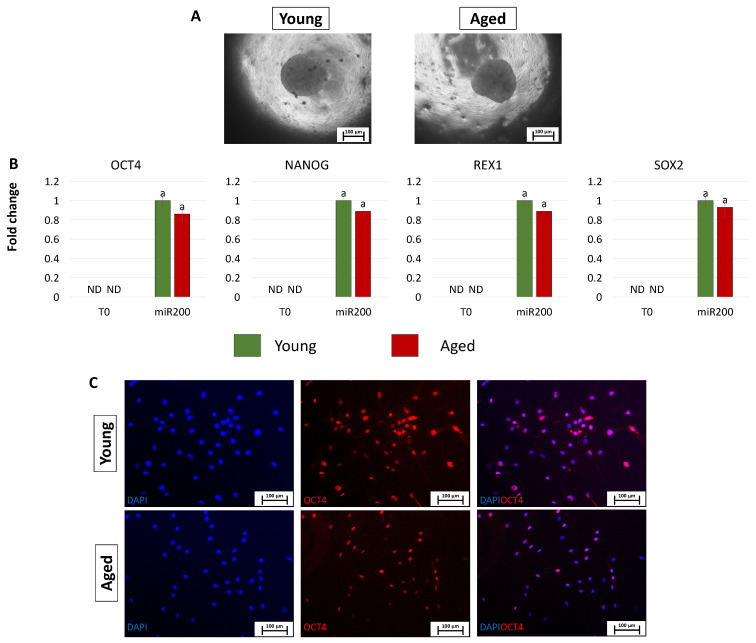
Creation of ICM-like spheroids from miR-200-reprogrammed cells: (**A**) Young and aged fibroblasts transfected with miR-200 and encapsulated in PTFE micro-bioreactors form 3D spherical structures (scale bars: 100 μm). (**B**) Expression levels for the pluripotent-related genes OCT4, NANOG, REX1, and SOX2 in young (green bars) and aged (red bars) untreated fibroblasts (T0) and in fibroblasts transfected with miR-200 and encapsulated in PTFE micro-bioreactors (miR.200). Gene transcription values are reported with the highest expression set to 1 and all others relative to this. Different superscripts denote significant differences (*p*  <  0.05). (**C**) Immunostainings show immunopositivity of cells isolated from young and aged ICMs for the pluripotency-related marker OCT4. Nuclei are stained blue (scale bars: 100 μm).

**Figure 5 cells-13-00628-f005:**
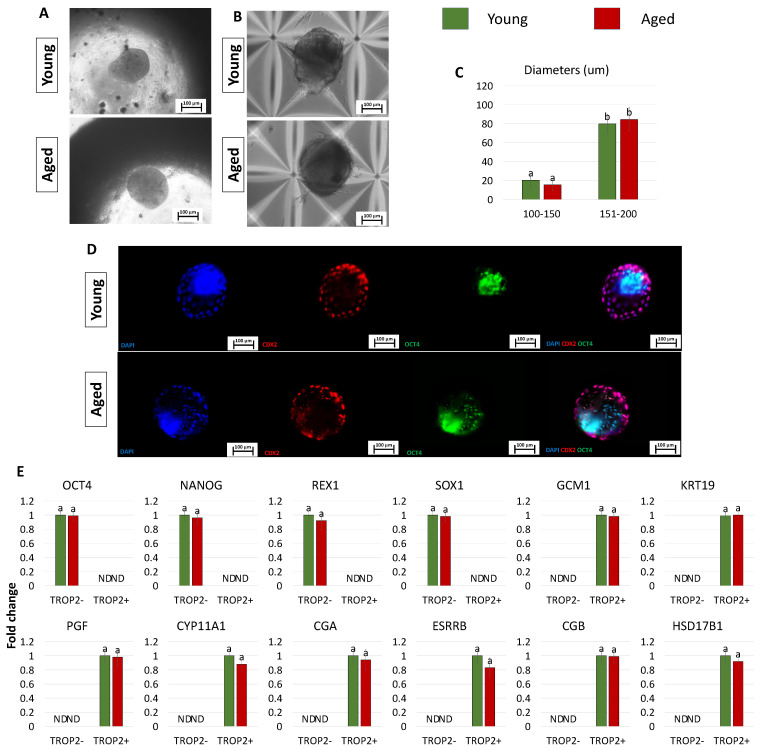
Generation of blastoids by assembling TR-like cells and ICM-like spheroids obtained from miR-200-reprogrammed cells: (**A**,**B**) Representative images of young and aged blastoids cultured in PTFE ((**A**) scale bars: 100 μm) and in micro-wells ((**B**) scale bars: 100 μm). (**C**) Rates of young (green bars) and aged (red bars) blastoids showing diameters ranging from 100 to 150 μm and from 151 to 200 μm. Superscripts ^a,b^ indicate significant differences (*p*  <  0.05). (**D**) Representative immunostaining of young and aged blastoids showing CDX2+ cells (red) localized to the outer layer of the spheroids and OCT4+ cells (green) in the inner compartment. Nuclei are stained blue (scale bars: 100 μm). (**E**) Transcription levels for pluripotency- (OCT4, NANOG, REX1, SOX2) and TR-related genes (GCM1, KRT19, PGF, CYP11A1, CGA, ESRRB, CGB, HSD17B1) in TROP2+ and TROP2- young (green bars) and aged (red bars) cells. Gene transcription is indicated with the highest expression set to 1 and all others relative to this.

**Table 1 cells-13-00628-t001:** List of primers used for quantitative PCR analysis.

GENE	DESCRIPTION	CAT.N.
*ACTB*	Actin, beta	Hs01060665_g1
*CGA*	Glycoprotein hormones, alpha polypeptide	Hs00985275_g1
*CGB*	Chorionic gonadotropin beta	Hs03407524_uH
*CYP11A1*	Cytochrome P450 family 11, subfamily A, member 1	Hs00167984_m1
*ESRRB*	Estrogen-related receptor beta	Hs01584024_m1
*GAPDH*	Glyceraldehyde-3-phosphate dehydrogenase	Hs02786624_g1
*GCM1*	Glial cells missing homolog 1	Hs00172692_m1
*HSD17B1*	Hydroxysteroid 17-beta dehydrogenase 1	Hs00166219_g1
*KRT19*	Keratin 19	Hs00761767_s1
*NANOG*	Nanog homeobox	Hs02387400_g1
*OCT4*	POU class 5 homeobox 1	Hs04260367_gH
*PGF*	Placental growth factor	Hs00182176_m1
*REX1*	ZFP42 zinc finger protein	Hs01938187_s1
*SOX2*	SRY-Box transcription factor 2	Hs04234836_s1

## Data Availability

The data presented in this study are available on request from the corresponding author.

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
