# Peer review of "Generation of Artificial Blastoids Combining miR-200-Mediated Reprogramming and Mechanical Cues"

_cells, 2024, doi:10.3390/cells13070628_

Round 1

Reviewer 1 Report

Comments and Suggestions for Authors

In the manuscript (Cells-2883818) titled “Generation of artificial blastoids combining miR-200-mediated 2 reprogramming and mechanical cues,” authors report a protocol to generate human blastoids from adult fibroblasts using miR-200b/c-based cellular reprogramming and Polytetrafluoroethylene (PTFE) encapsulation of the reprogrammed cells. The authors used synthetic miR-200b/c  to reprogram human skin fibroblasts, which they obtained from the Coriell Institute, and classified them as young (collected from individuals aged 29-32 years) and old (collected from individuals aged 82-96 years). In their results, the authors showed that miR-200b/c transfected fibroblasts developed distinguishable cellular aggregates, which expressed pluripotency genes. The reprogrammed cells were then used to differentiate trophectoderm-like cells and to generate reprogrammed cells 3D spheroids with PTFE encapsulation, which the authors referred to as ICM-like spheroids. The generated 3D ICM-like spheroid and trophectoderm-like cells were then assembled into a spheroid and encapsulated in PTFE using a method described in their previous publication. The generated spheroids/blastoids were imaged on day 2 and day 7 of the culture (Fig. 4-A & 4-B). Finally, the generated blastoids were dissociated into single cells, and the cells were then sorted based on the trophectoderm cell surface marker (TROP2) and analyzed for the expression of pluripotency-related and trophectoderm-related genes. As expected, the results show that TROP2 positive cells expressed trophectoderm genes and TROP2 negative cells expressed pluripotency-related genes. This reviewer has significant concerns regarding the results and conclusion of this manuscript that the author needs to address before the manuscript can be considered for publication.

Major concerns:

1.       The results fail to support the conclusion that the authors were successful in generating human blastoids. The gene expression data from dissociated cells does not provide any evidence of the cellular organization in the generated blastoids. The trophectoderm cells and the pluripotent cells were already expressing the markers the authors analyzed post-blastoid assembly. Furthermore, from Figure 4-B it is not clear if the morphology of generated spheroids is consistent with the blastoid morphology. To support their conclusion the authors must include data that shows the localization of trophectoderm and pluripotent cells in the generated blastoids.

2.       The manuscript does not provide any significant improvement to the existing blastoid generation protocols or any new insight or data on the generated blastoids/spheroids. 

3.       The overall characterization of the generated cells and the blastoids was limited to very few markers and mostly relied on relative quantification of gene expression. Also, the results were presented as mean/average per category. Were cells from multiple blastoids analyzed separately? Was there variation from one blastoid to another?

4.       The fibroblast samples used in the study were divided into two categories, “Young” and “Aged,” and all the results were presented relative to these categories. However, the authors did not provide any explanations or the significance of this categorization to the blastoid generation.

5.       The authors also categorized the generated blastoids into two sizes i.e. 151 – 200 µm and 100 – 150 µm. What was the significance of these blastoid size categories?

6.       The authors indicated that their approach “allows for an efficient in vitro generation of artificial blastoids”. However, they did not provide any data on the efficiency of their protocol.

Minor concerns:

1.       The authors described their protocol as a novel strategy. However, neither their cellular reprogramming with microRNA nor their strategy of blastoid generation is novel.

2.       There is significant linguistic ambiguity throughout the manuscript.  

3.       There is excessive self-citation of the author's previous publications without much relevance to this manuscript. For example, references 14 – 19.   

Overall, in this reviewer’s opinion, the authors should include data that shows the localization of trophectoderm and pluripotent cells in the generated blastoids. The authors should also perform a more in-depth analysis of the generated spheroids/blastoids, particularly providing evidence that their method is more efficient than the methods previously published.

Comments on the Quality of English Language

There is significant linguistic ambiguity throughout the manuscript.  

Author Response

In the manuscript (Cells-2883818) titled “Generation of artificial blastoids combining miR-200-mediated 2 reprogramming and mechanical cues,” authors report a protocol to generate human blastoids from adult fibroblasts using miR-200b/c-based cellular reprogramming and Polytetrafluoroethylene (PTFE) encapsulation of the reprogrammed cells. The authors used synthetic miR-200b/c  to reprogram human skin fibroblasts, which they obtained from the Coriell Institute, and classified them as young (collected from individuals aged 29-32 years) and old (collected from individuals aged 82-96 years). In their results, the authors showed that miR-200b/c transfected fibroblasts developed distinguishable cellular aggregates, which expressed pluripotency genes. The reprogrammed cells were then used to differentiate trophectoderm-like cells and to generate reprogrammed cells 3D spheroids with PTFE encapsulation, which the authors referred to as ICM-like spheroids. The generated 3D ICM-like spheroid and trophectoderm-like cells were then assembled into a spheroid and encapsulated in PTFE using a method described in their previous publication. The generated spheroids/blastoids were imaged on day 2 and day 7 of the culture (Fig. 4-A & 4-B). Finally, the generated blastoids were dissociated into single cells, and the cells were then sorted based on the trophectoderm cell surface marker (TROP2) and analyzed for the expression of pluripotency-related and trophectoderm-related genes. As expected, the results show that TROP2 positive cells expressed trophectoderm genes and TROP2 negative cells expressed pluripotency-related genes. This reviewer has significant concerns regarding the results and conclusion of this manuscript that the author needs to address before the manuscript can be considered for publication.

We thank the Reviewer for her/his comments that will help our manuscript reviewing process.

Major concerns:

  1. The results fail to support the conclusion that the authors were successful in generating human blastoids. The gene expression data from dissociated cells does not provide any evidence of the cellular organization in the generated blastoids. The trophectoderm cells and the pluripotent cells were already expressing the markers the authors analyzed post-blastoid assembly. Furthermore, from Figure 4-B it is not clear if the morphology of generated spheroids is consistent with the blastoid morphology. To support their conclusion the authors must include data that shows the localization of trophectoderm and pluripotent cells in the generated blastoids.

As requested by the Reviewer images showing the localization of trophectoderm and pluripotent cells in young and aged blastoids were included in the new version of Figure 5. Material and methods, Results, figure legend 5 and discussion were modified accordingly. Please see lines 293-295 (results: “Immunostaining studies showed CDX2+ cells localized to the outer layer of the generated blastoids, while OCT4 was expressed by cells of the inner compartment (Figure 5D).”), Figure 5, lines 313-316 (figure legend: “(D) Representative immunostaining of young and aged blastoids showing CDX2+ cells (red) localize to the outer layer of the spheroids and OCT4+ cells (green) in the inner compartment. Nuclei are stained in blue (scale bars 100 μm)”) and lines 378-383 (discussion: “In addition, the localization of cells within the spheroids, demonstrated by the immunostaining studies showing CDX2+ cells localized externally, surrounding the generated blastoids, and OCT4+ cells closely assembled within them, further indicate that the newly formed spheroids display a cell lineage segregation similar to natural blastocysts, and that this feature is maintained both in young as well as in aged blastoids.”).

  1. The manuscript does not provide any significant improvement to the existing blastoid generation protocols or any new insight or data on the generated blastoids/spheroids.  

We believe that the combined use of miR-200 with mechanical cues is a novel approach as also stated by Reviewer #2. In addition, to our knowledge, the generation of blastoid from senescent cells has not been reported before.

  1. The overall characterization of the generated cells and the blastoids was limited to very few markers and mostly relied on relative quantification of gene expression. Also, the results were presented as mean/average per category. Were cells from multiple blastoids analyzed separately? Was there variation from one blastoid to another?

Cell analyses were carried out per single blastoid. Variation from one blastoid to another exists and the relative value can be appreciated as standard deviation (SD) in each histogram of Figure 5D.

A sentence in the material and methods section was added to clarify this point (please see line165 “Single blastoids were separately analyzed. More in detail, each blastoid was dissociated to single cell suspension“).

  1. The fibroblast samples used in the study were divided into two categories, “Young” and “Aged,” and all the results were presented relative to these categories. However, the authors did not provide any explanations or the significance of this categorization to the blastoid generation.

In a previously published paper, we could detect miR-200 ability to ameliorate cellular and physiological hallmarks of aging in senescent fibroblasts, that, however, maintained their phenotype, thus undergoing “partial reprogramming” also defined as “age reprogramming”. Based on this observation, in the present manuscript, we used the same categories (“Young” and “Aged”), but we investigated whether the combination of miR-200 with mechanical cues (PTFE microbioreactors) would allow successful “developmental reprogramming”, with cell de-differentiation to a high plasticity state and the generation of blastoid structures.

A sentence was added to the introduction section to better explain this aspect. Please see lines 59-67 “In addition, it has been recently demonstrated that the use of miR-200s is able to ameliorate cellular and physiological hallmarks of aging in senescent cells [21,22], that, however, maintained their phenotype, thus undergoing “partial reprogramming” al-so defined as “age reprogramming” [28]. Based on this observation, in the present work, we investigate whether the combination of miR-200 with mechanical cues would allow successful “developmental reprogramming”, with cell de-differentiation to a high plasticity state and the generation of blastoid structures, starting from easily accessible terminally differentiated fibroblasts isolated from young and aged individuals (Figure 1).”).

  1. The authors also categorized the generated blastoids into two sizes i.e. 151 – 200 µm and 100 – 150 µm. What was the significance of these blastoid size categories?

We categorized the generated blastoids into spheroids exhibiting diameters ranging from 151 – 200 µm and 100 – 150 µm, based on the average reported values of natural blastocysts that measure 175–211 μm and in agreement with the parameters and criteria currently used to define human blastoid models (Kagawa et al., Nature 2022;601(7894):600-605.  doi: 10.1038/s41586-021-04267-8).

We think this was also stated in the discussion section lines 373-377.

  1. The authors indicated that their approach “allows for an efficient in vitro generation of artificial blastoids”. However, they did not provide any data on the efficiency of their protocol.

Around 80% of the generated spheroid structures exhibited all the accepted parameters and criteria used to define blastocyst-like in vitro models, namely timing of development, diameters, cell lineage differentiation and appropriate spatial compartmentalization. This leads us to believe that our approach is efficient.

Minor concerns:

  1. The authors described their protocol as a novel strategy. However, neither their cellular reprogramming with microRNA nor their strategy of blastoid generation is novel.

We agree with the observation that neither cellular reprogramming with microRNA nor strategy of blastoid generation is novel. However, we believe that the combined use of miR-200s with mechanical cues is a novel approach, as also stated by Reviewer #2. In addition, to our knowledge, the generation of blastoid from senescent cells has not been reported before.

  1. There is significant linguistic ambiguity throughout the manuscript.  

We edited language throughout the manuscript.

  1. There is excessive self-citation of the author's previous publications without much relevance to this manuscript. For example, references 14 – 19.   

We would like to draw the Reviewer’s attention that references 14 – 19 are, in our opinion, essential to introduce and justify the present experiments, since they constitute the background evidence that have brought us to the protocol described in this manuscript.

Overall, in this reviewer’s opinion, the authors should include data that shows the localization of trophectoderm and pluripotent cells in the generated blastoids. The authors should also perform a more in-depth analysis of the generated spheroids/blastoids, particularly providing evidence that their method is more efficient than the methods previously published.

Please see above (response 1). Also note that we never compared the efficiency of our protocol with that of the methods previously published, also because no previous data are available on blastoid obtained from aged cells.

Reviewer 2 Report

Comments and Suggestions for Authors

The manuscript “Generation of artificial blastoids combining miR-200-mediated 2 reprogramming and mechanical cues” by Pennarossa et al. presents a novel strategy for creating artificial blastoids by exploting the capabilities of miR-200 family. The reprogrammed dermal fibroblasts are induced towards the trophoectoderm lineage and inner-cell mass like spheroids. The obtained cell types are then co-cultured and directed towards blastoid formation. The phenotype and genotype of the cell lineages are supported by immunocytochemical staining images, gene expression analysis and bright-field images. However, it can be improved to arouse the interest of readers towards the research by complying with the following suggestions.

Minor Comments:

1.     It would be nice to see an experimental timeline indicating the stages of the experiment along with the number of days to better understand the steps of the re-programming and culture process.

2.     Cavitation is a crucial aspect of blastoid formation. Have the authors validated the cavitation formation in the generated blastoids?

3.     It is clear from the list of materials that the authors have not used ROCK inhibitor in their stem cell culture media. How did they prevent cell apoptosis during the complex process of forming TR lineage and ICM-like spheroids? Have they used any alternative for the same?

4.     Future aspect of the developed protocol or blastoids is not explained. A few lines or a short paragraph about the future perspectives will be appreciated.

Author Response

The manuscript “Generation of artificial blastoids combining miR-200-mediated 2 reprogramming and mechanical cues” by Pennarossa et al. presents a novel strategy for creating artificial blastoids by exploting the capabilities of miR-200 family. The reprogrammed dermal fibroblasts are induced towards the trophoectoderm lineage and inner-cell mass like spheroids. The obtained cell types are then co-cultured and directed towards blastoid formation. The phenotype and genotype of the cell lineages are supported by immunocytochemical staining images, gene expression analysis and bright-field images. However, it can be improved to arouse the interest of readers towards the research by complying with the following suggestions.

We thank the Reviewer for her/his constructive suggestions that will improve the quality of the manuscript.

Minor Comments:

  1. It would be nice to see an experimental timeline indicating the stages of the experiment along with the number of days to better understand the steps of the re-programming and culture process.

An experimental timeline is now included in the new version of the manuscript. Please see Figure 1.

  1. Cavitation is a crucial aspect of blastoid formation. Have the authors validated the cavitation formation in the generated blastoids?

Cavitation was evaluated in blastoids monitoring diameter changes along the culture period and considering “cavitated” all spheroids displaying a size >151 µm, with an evident fluid filled cavity (Lagalla et al., J Assist Reprod Genet. 2015; 32(5): 705–712 doi: 10.1007/s10815-015-0469-3).

  1. It is clear from the list of materials that the authors have not used ROCK inhibitor in their stem cell culture media. How did they prevent cell apoptosis during the complex process of forming TR lineage and ICM-like spheroids? Have they used any alternative for the same?

ROCK inhibitor was not used in our stem cell culture medium, since cells are grown on gelatin matrix and mechanically detached from it, avoiding any single cell dissociation protocol which is the main indication for ROCK inhibitor use. In addition, we think that ROCK inhibitor should be included in the culture medium only when strictly required, since it has been reported to affect SOX2 expression, limiting pluripotency and priming cells for EMT (Narva et al., Stem Cell Reports 2017;9(1):67-76.  doi: 10.1016/j.stemcr.2017.05.021) as well as to alter pluripotent cell metabolism (Vernardis et al., Sci Rep. 2017; 7: 42138. doi: 10.1038/srep42138).

  1. Future aspect of the developed protocol or blastoids is not explained. A few lines or a short paragraph about the future perspectives will be appreciated.

Thank you for this remark. A paragraph including the possible applications and future perspectives was added in the discussion section. Please see lines 393-395 “They will offer the advantages of scalability, accessibility, limited variables and direct manipulation. In future, these newly generated structures could be also useful to identify therapeutic targets as well as to support preclinical modelling, offering an ethical alter-native to the use of natural ones”.

Round 2

Reviewer 1 Report

Comments and Suggestions for Authors

In this revised version of the manuscript (cells-2883818-peer-review-r1), the authors addressed the major concern I had regarding the successful generation of human blastoids by adding new data in Figure 5. In the reader’s interest, I strongly suggest that the authors perform the following minor revisions before the publication of their manuscript.

Minor revisions:

1.  In the discussion section, please describe the significant improvement(s) the author’s protocol provides over the previously published human blastoid generation, in terms of novelty, efficiency, ease of performance, and/or improvement to the generated blastoids. Also, any novel insight from their analysis of the generated blastoids.

2.       Please address, apart from their size how were the blastoids of size 100 – 150 µm different from the 151 – 200 µm. If the average reported blastocyst size is 175–211 μm, what was the rationale for a 151 – 200 µm category/group?

3.       I am not convinced that all the 14-19 references are necessary for the background. I strongly suggest that the authors include references that are relevant to this study.

Comments on the Quality of English Language

Can be improved.

Author Response

In this revised version of the manuscript (cells-2883818-peer-review-r1), the authors addressed the major concern I had regarding the successful generation of human blastoids by adding new data in Figure 5. In the reader’s interest, I strongly suggest that the authors perform the following minor revisions before the publication of their manuscript.

We thank the Reviewer for her/his comments.

Minor revisions:

  1. In the discussion section, please describe the significant improvement(s) the author’s protocol provides over the previously published human blastoid generation, in terms of novelty, efficiency, ease of performance, and/or improvement to the generated blastoids. Also, any novel insight from their analysis of the generated blastoids.

As requested by the Reviewer, we added a paragraph in the discussion section. Please see lines 300-304: “To our knowledge, this strategy has not been applied before for the creation of blastocyst-like structures. Beside its novelty, the method offers the advantage of easily accessible cells as starting material and avoids the use of any retroviral and/or lentiviral vectors, as well as the insertion of transgenes. The significant improvement offered by the protocol described is also represented by the ease of performance that involves a simple transfection of miR-200s to induce a highly permissive state in adult dermal fibroblasts, isolated either from young or aged donors.”

  1. Please address, apart from their size how were the blastoids of size 100 – 150 µm different from the 151 – 200 µm. If the average reported blastocyst size is 175–211 μm, what was the rationale for a 151 – 200 µm category/group?

The rational for a 151-200 µm category/group agrees with the parameters and criteria currently accepted and used to define human blastoid models (Kagawa et al., Nature 2022;601(7894):600-605.  doi: 10.1038/s41586-021-04267-8; Fan Y et al., Generation of human blastocyst-like structures from pluripotent stem cells. Cell Discov. 2021 Sep 7;7(1):81. doi: 10.1038/s41421-021-00316-8; Kagawa H et al, Protocol for Human Blastoids Modeling Blastocyst Development and Implantation. J Vis Exp. 2022 Aug 10;(186). doi: 10.3791/63388; Longsdon et al., Optimizing extended culture conditions of stem cell derived blastoids to mimic human implantation in vitro. Fertility and Sterility Oct 2022, doi.org/10.1016/j.fertnstert.2022.08.123).

No further analyses were carried out on spheroids of size 100 – 150 µm, because, due to their suboptimal diameters, they were not considered as “blastoids”. However, we agree with the Reviewer that this is an interesting aspect that could be investigated in future studies.

  1. I am not convinced that all the 14-19 references are necessary for the background. I strongly suggest that the authors include references that are relevant to this study.

As strongly requested by the Reviewer, we removed the references that are not indispensable to support the method described in the present study. However, we would like to stress that the removed references, although dispensable, were, in our opinion, relevant to contextualize the background research that led to the reprogramming strategy proposed in this manuscript.